# A Review of the Intrinsic Chemical Stability Challenge in Operational Perovskite Photovoltaics

**DOI:** 10.3390/ma18204776

**Published:** 2025-10-19

**Authors:** Huān Bì, Zhen Wang, Zhenhua Xu

**Affiliations:** 1Info-Powered Energy System Research Center (i-PERC), The University of Electro-Communications, Tokyo 182-8585, Japan; 2Key Laboratory of Flexible Electronics (KLOFE) & Institute of Advanced Materials (IAM), Nanjing Tech University (NanjingTech), 30 South Puzhu Road, Nanjing 211816, China; 3School of Materials Science and Engineering, NingboTech University, Ningbo 315100, China

**Keywords:** perovskite photovoltaic, intrinsic chemical stability, deprotonation of methylammonium cations, halide ions migration, redox of halide ions

## Abstract

Although the power conversion efficiency of perovskite photovoltaics (PVs) has achieved significant progress, the operational stability is still a critical issue for their commercialization. Compared to inorganic semiconductor materials, organic species in perovskites are intrinsically unstable under long-term illumination, heat, and bias stresses. These organic species exhibit higher chemical reactivity, which can complicate the degradation mechanisms or model of perovskite PVs. In this review, we analyzed the types of chemical reactions for different organic species. The chemical instability mainly stems from the deprotonation of A-site ammonium cations, X-site halide ion migration, and oxidation of halide ions, which can even mutually influence one another. We systematically discuss the effect of this chemical instability on perovskite structure degradation under device operation. These special chemical evolutions will accelerate perovskite PVs’ degradation. Then, strategies to mitigate these reactions for enhanced operational stability are introduced. Despite substantial progress in the operational stability of perovskite PVs, achieving an operational lifetime comparable to crystalline silicon remains challenging. Therefore, a deep understanding of intrinsic perovskite structure degradation should become a research focus, contributing to improvement in the operational lifetime of perovskite PVs.

## 1. Introduction

The power conversion efficiency of perovskite photovoltaics (PVs) is already competitive with that of crystalline silicon photovoltaics [1]. This high efficiency stems from additive or interface engineering strategies, which can reduce defects for the suppression of nonradiative recombination and high carrier transport [2,3,4]. These defects’ suppression is reported to achieve long operational stability through preventing perovskite lattice degradation initiated by numerous defects [5,6,7]. However, the lifetime of these new photovoltaic technologies lags far behind that of silicon-wafer-based solar cells by 20–25 years, and thus they do not fulfill the minimum operational stability for commercialization [8,9]. Previous work has demonstrated that perovskite materials, carrier transport materials, and electrode materials all easily degrade under external stresses, collectively inducing a rapid efficiency drop in perovskite PVs or modules [2,10]. Extensive experience from other PVs can provide valuable guidance for stabilizing carrier transport and electrode materials in perovskite PVs. However, organic–inorganic lead halide perovskites show a different degradation mechanism compared to inorganic photoactive material-based PVs. Differing from silicon photovoltaics, the reactivity of organic components in perovskites is detrimental to the stability of perovskite PVs under external stresses [11,12,13]. Despite significant advances in perovskite photovoltaics, a fundamental understanding of the chemical instability and degradation mechanisms underlying methods for enhancing the stability of organic components remains insufficiently explored.

In this review, we focus on the structural and chemical evolution process in operational perovskite PVs. First, we summarize the current operational stability development of perovskite PVs. Then, we systematically discuss the chemical degradation mechanism of the perovskite structure. Organic species of A-site ammonium cations and X-site halide ions exhibit higher chemical reactivity under continuous illumination, heat, and bias stresses. Finally, we outline strategies for enhancing the operational stability of perovskite PVs according to the chemical degradation mechanisms of perovskite structures.

## 2. Status of Perovskite PV Stability

In recent years, extensive efforts have focused on the investigation of degradation mechanisms through testing operational stability in perovskite PVs. Table 1 summarizes the operational lifetime of a perovskite PV under a given initial device efficiency. We did not collect the operational stability data based on probable low-efficiency perovskite PVs because such results presumably cannot reflect the degradation mechanisms of practical high-performance devices. We note that the lifetime of most devices is in the range of 1000~2000 h, regardless of the device’s architecture (n-i-p or p-i-n). Although substantial progress has been made in improving the stability of perovskite PVs, their operational lifetime remains far shorter than that of crystalline silicon PVs, which typically exceeds 20 years (>175,000 h). Reported perovskite PVs typically sustain an operational stability of only a few thousand hours under laboratory conditions, highlighting a considerable gap in meeting commercialization requirements. The shortened lifetime of perovskite PVs arises from multiple factors involving the degradation of charge transport materials, photoactive materials, and electrode materials. Crystalline silicon materials exhibit exceptional long-term stability due to the robustness of their inorganic silicon atoms and strong covalent bonding. In contrast, perovskite structures often undergo severe degradation within only a few thousand hours. Therefore, particular attention should be firstly directed toward enhancing the intrinsic stability of the photoactive perovskite materials. We found that formamidinium–cesium mixed-cation composition (FAC) perovskites exhibit higher stability compared to methylammonium (MA)-containing perovskites. This phenomenon is presumably due to the deprotonation of methylammonium into MA^0^, which leads to chemical interactions with other cations [14]. Although the perovskite PV has achieved > 27% efficiency, the device’s lifetime is tested based on relatively low device performance (Table 1). We suspect that the reduced recombination and enhanced carrier transport in higher-efficiency perovskite PVs can induce device vulnerability to ion migration. Therefore, operational lifetime testing in higher-efficiency perovskite PV modules is necessary for commercialization.

## 3. Chemical Evolution of Perovskites

Traditional inorganic semiconductors have covalently stable bonding, contributing to the long-term durability of these material-based optoelectronic devices. In contrast, perovskites are very different from inorganic semiconductor materials in two key aspects. First, the perovskite lattice is held together by ionic and hydrogen bonding. This fragile bonding can result in pronounced ion migration, rendering the perovskite lattice similar to fluids. Second, organic compositions are chemically active, meaning they easily experience chemical reactions under continuous illumination or thermal stresses. These reactions can generate vacancy defects, accelerating ion migration through the vacancy channel and ultimately leading to the structural collapse of perovskites. In organic–inorganic ABX_3_ perovskites, the organic components of A and X sites exhibit higher chemical activity, in which the A site is FA^+^/MA^+^ and the X site is I^−^ (iodide). First, A site-related reactions are often initiated by the deprotonation of ammonium cations. Second, the redox of iodide ions can induce the formation of iodide vacancy-related defects, which accelerate ion migration and thus lead to irreversible degradation of operational perovskite PVs. In this review, the literature on chemical reactions of lead halide perovskites is selected to comprehensively discuss intrinsic stability during the operation of perovskite PVs. Most of these studies were published after 2018 to ensure that the discussion can reflect recent and relevant advances in perovskite PV stability research.

### 3.1. Deprotonation of A-Site Ammonium Cations

FA-rich perovskite has been widely employed for high-stability PVs compared to MA-containing perovskites. The advantages of FA cations arise from two main factors. First, FA cations have an N-C-N backbone with delocalized resonance structures, enhancing their intrinsic stability (Figure 1a) [29]. Second, four N-H bonds in FA cations can form multiple hydrogen bonds with lead octahedra, further stabilizing the perovskite lattice [10]. We should mention that MACl is a prerequisite for manipulating perovskite crystallization for improved efficiency of FA-based perovskite PVs. However, MACl can react with FA^+^ to form MFA^+^ (methyl formamidinium) or DMFA^+^ (dimethyl formamidinium) (Figure 1b) [14]. Specifically, deprotonation of MA^+^ into MA^0^ reversibly triggers an addition–elimination reaction between MA^0^ and FA^+^ under high-temperature annealing, which results in the formation of MFA^+^ products [14]. Cis-MFAI isomers can react with PbI_2_ to form one-dimensional face-sharing octahedron structures of 2H-MFAPbI_3_ (Figure 1c), which are similar to the 2H-FAPbI_3_ structure (δ-FAPbI_3_) [30]. Therefore, the existence of the 2H-MFAPbI_3_ phase is not beneficial for the carrier transport and long-term stability of perovskite PVs. In order to avoid these side products, Yuan et al. replaced MACl with CsCl for perovskite crystallization manipulation, leading to >26% efficiency of FAC-based perovskite PVs [16].

The FA^+^ itself can also deprotonate at the interface of metal oxides (as charge transport layers)/perovskites under external light/thermal stresses [31]. These metal oxides, including NiO_X_, ZnO, SnO_2_ and TiO_2_, all show reactivity with FA^+^ in perovskites through acid–base reactions [24]. This interfacial deprotonation is attributed to strong proton adsorption capability with enhanced interfacial proton transfer, which probably stems from the high fracture energy at the device interface (Figure 1d) [19]. As a result, I^−^ is left to form a PbI_2_ sheet after the deprotonation of FAI at the device interface. This proton transfer can be accelerated by higher temperatures or continuous light soaking (Figure 1e) [19,32]. In addition, the deprotonation of FA^+^ also induces the irreversible formation of 1,3,5-triazine through a condensation reaction under higher temperature [33].

In addition to small-sized FA^+^/MA^+^, large ammonium-terminated cations can also undergo diverse chemical evolutions during device operation. Large cations are employed to form 2D perovskite on an FA-based 3D perovskite film surface. This 2D capping layer can not only reduce defects on the perovskite surface or at the perovskite grain boundaries (GBs) but also hinder moisture ingression. Phenethylammonium (PEA^+^) and butylammonium (BA^+^) are the most commonly used ammonium cations. However, studies show that these large ammonium cations exert poor thermal stability for perovskite PVs (Figure 1f) [25,34,35,36,37]. Huang et al. found that the easy deprotonation of PEA^+^/BA^+^ can destroy the 2D perovskite capping layer due to the low *p*Ka value of these ammonium cations [37]. As a result, an addition–elimination reaction between PEA^0^/BA^0^ and FA^+^ occurs, causing the formation of FA vacancy defects, which leads to the accelerated degradation of perovskite PVs. We can note that a high *p*Ka value of large ammonium-terminated cations is required to resist this deprotonation process (Figure 1g), which can be achieved by enhancing the degree of molecular conjugation [10,37,38]. Therefore, a promising approach is to choose amidinium- or benzene-based ammonium-terminated cations. Amidinium-terminated cations with an N-C-N planar structure have favorable resonance stabilization, which can efficiently suppress deprotonation for highly stable perovskite PVs [39,40]. For instance, Park et al. demonstrated that 2-amidinopyridine cations with a conjugated delocalized system have higher structural ordering on the perovskite surface, exhibiting enhanced passivation effects for more stable perovskite PVs [41].

**Figure 1 materials-18-04776-f001:**
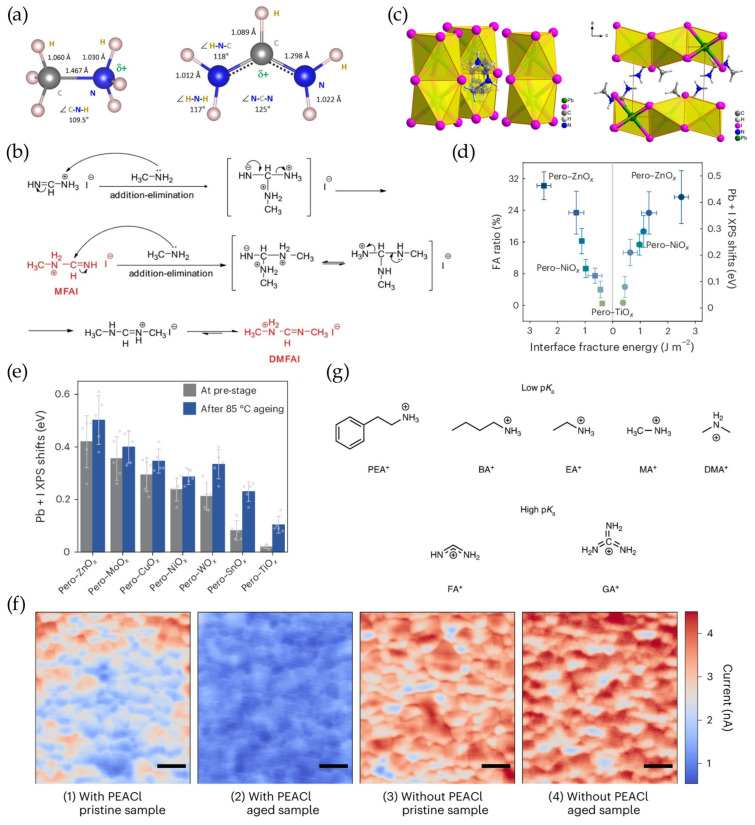
(**a**) Structural formulas of methylammonium and formamidinium cations, reprinted with permission from Ref. [29]. (**b**) The condensation reactions between deprotonated MA^0^ and FA^+^, which lead to MAF^+^ and DMFA^+^, reprinted with permission from Ref. [14]. (**c**) Crystal structure of a single yellow 2H-MFAPbI_3_ crystal with disordered and ordered cis-MFA cations, reprinted with permission from Ref. [30]. (**d**) The correlation between the interface fracture energy and the interface proton transfer interaction intensity. As the interfacial fracture energy increases, the ratio of FA^+^ participating in interfacial transfer and the XPS peak shift in Pb + I all increase. (**e**) The sum of the Pb + I XPS shifts before and after 85 °C treatment. The perovskite degradation becomes more pronounced as the interfacial fracture energy increases. This was reprinted with permission from Ref. [19]. (**f**) Electron beam-induced current (EBIC) mapping of perovskites with or without PEACl before and after 85 °C aging under 1-sun illumination for 250 h. Incorporation of PEA^+^ can induce lower and less uniform current in EBIC mapping. The scale bar is 2 μm. (**g**) Chemical structure of common ammonium cations. Ammonium cations with higher *p*Ka levels contribute to enhanced perovskite stability. This was reprinted with permission from Ref. [37].

### 3.2. Mobile X-Site Halide Ion Migration

Numerous studies have shown that halide vacancies (V_I_) and interstitial iodide ions (I_i_) are the dominant mobile ions undergoing migration as a result of having the lowest activation energy among the various ions in perovskites [42,43,44,45]. Frenkel defect pairs—V_I_ and I_i_—are easily formed because of the weaker ionic bonds of Pb-I in “fluid” perovskites [46]. This straight-forward defect formation results in a higher mobile ion concentration, which ranges from 1 × 10^15^ to 1 × 10^17^ cm^−3^ in polycrystalline perovskite films [47,48]. Although these defects are shallow and generally benign, their low activation energy induces a higher halide ion migration rate. The activation energy of ion migration is influenced by the A site composition in perovskites. Studies show that FA^+^ has stronger hydrogen bonds with halide ions compared to MA^+^, leading to a higher ion migration activation energy [49]. Inorganic Rb (rubidium) or Cs (Cesium) ions at the A site can induce lattice distortion, also increasing the activation energy barriers [50]. For example, Yuan et al. found that the presence of 5% Cs^+^ at the A site can induce a constriction of the PbI_6_ octahedral structure, which compresses the channels of ion migration [51]. The properties of solvents in a perovskite precursor solution can also suppress ion migration through regulating perovskite crystallization with fewer defects [52]. The N-methyl-2-pyrrolidone (NMP) solvent can strongly interact with FAI/PbI_2_ compared to the dimethylformamide (DMF) solvent, improving the quality of the perovskite thin films with reduced interfacial halide vacancies [53]. This vacancy-mediated ion hopping will accelerate ion migration at the interface, leading to the formation of more vacancy defects. In addition, I_i_ defects can lower the activation energy of the α-FAPbI_3_-to-δ-FAPbI_3_ phase transformation, accelerating the degradation of the cubic perovskite phase during device operation (Figure 2a) [54].

Although halide ion migration has been recognized as a key factor for rapid efficiency loss, the effect of mobile halide ions on device degradation parameters has not been thoroughly studied. Huang et al. found that electron-only wide-bandgap perovskite PVs exhibit ion migration-induced phase segregation, whereas this phenomenon is not observed in hole-only devices [55]. This observation suggests that ion migration is induced by the excess electron, instead of the excess hole (Figure 2b) [55]. The energy barrier of ion migration actually originates from the Coulombic interaction between mobile ions and V_I_. However, the positively charged V_I_ is neutralized by a photogenerated electron, inducing reduced interactions between mobile ions and V_I_, which promotes severe ion migration (Figure 2c). This ion migration can be suppressed by mixing cation–anion compositions in perovskites. Cations and anions of different sizes can induce local distortion in the crystal lattice, requiring more energy for ion migration within the lattice [56]. However, perovskites with mixed compositions are prone to phase segregation [57,58]. To further quantify the effect of mobile ions on device performance degradation, bias-assisted charge extraction (BACE) has been employed to determine the mobile ion concentration (Figure 2d) [59,60,61]. A voltage close to the open-circuit voltage is applied for several seconds in perovskite PVs before stitching to 0 V. Then the integrated transient current is equal to the mobile ion concentration. However, the BACE method only approximates the ion concentration. When the mobile ion concentration increases, more potential drops occur at the interface of the perovskite layer and the carrier transport layers, leading to only partial contribution of the ions to the current [62]. Stolterfoht et al. demonstrated that device performance degradation primarily stems from the loss of *J*_sc_ and FF, which is due to the field screening effect upon mobile ion-induced migration (Figure 2e) [63]. Furthermore, ion migration becomes more severe as the device ages, which leads to a higher concentration of mobile ions, measured through BACE (Figure 2f). Interestingly, almost no change in the efficiency drop at the initial aging timepoint was exhibited, followed by a linear increase in device efficiency loss (Figure 2g).

**Figure 2 materials-18-04776-f002:**
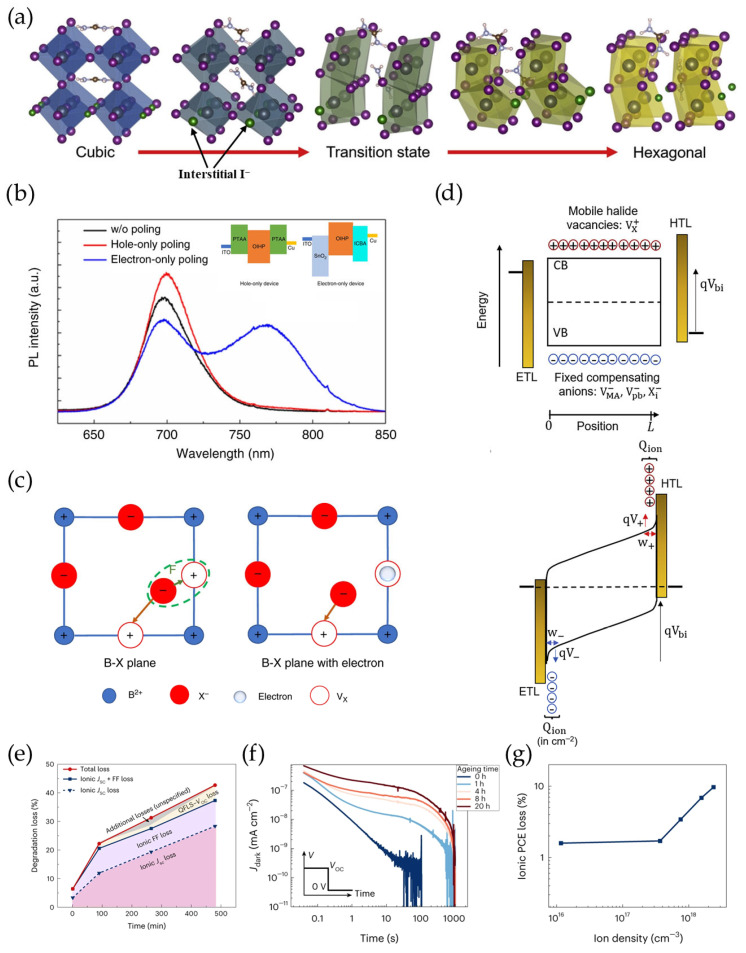
(**a**) Phase evolution with iodine interstitials, reprinted with permission from Ref. [54]. (**b**) PL spectra of hole-only and electron-only devices based on FA_0.85_Cs_0.15_Pb(I_0.6_Br_0.4_)_3_ under an electric field of 3 V μm^−1^ for 60 min. (**c**) Proposed mechanism for the effect of charge on halide ion migration and perovskite stability. V_I_ can coordinate with electrons, leading to reduced interactions between V_I_ and mobile ions. Then mobile ions undergo severe migration. This was reprinted with permission from Ref. [55]. (**d**) Schematic energy diagrams of the uniform distribution of mobile ions and accumulation of mobile cations/anions at the interface, reprinted with permission from Ref. [61]. (**e**) Contribution to device performance degradation by *J*_sc_, FF and *V*_oc_. (**f**) Current transients from BACE measurements after various aging times, where the voltage pulse applied to the device can be seen in the inset. The integral of the current is nearly equal to the mobile ion concentration. (**g**) The device efficiency degradation as a function of the mobile ion concentration. The ions concentration is calculated from the integral of the current from Figure 2f. This was reprinted with permission from Ref. [63].

### 3.3. Oxidation of X-Site Halide Ions

Although the above-mentioned mobile ion migration is reversible, the possibility of halide ion reactivity increases during ion migration. This behavior is analogous to the electrochemical iodide/triiodide redox in dye-sensitized solar cells (DSSCs), highlighting the high reactivity of ionic iodide [64]. As expected, the redox of halide ions in perovskites is detrimental to the long-term operational stability of perovskite PVs. I^−^ is readily oxidized into iodide (I_2_) molecules in operational perovskite PVs. Specifically, ammonium cations deprotonate into amine and hydrogen iodide (HI). Subsequently, HI undergoes a redox reaction to form I_2_ under ambient conditions (Figure 3a) [65,66,67]. PbI_2_ can also decompose into Pb^0^ and I_2_ under light illumination in operational perovskite PVs [68,69]. This redox of halide ions into I_2_ is presumably initiated by the light and bias, where free carriers are generated [63].

The formation of iodide-related species is complex and includes I* radicals, I_2_ and I_x_^−^ (known as polyiodides) under continuous illumination [67]. These species are mobile compared to the halide from the Pb-I bond, which will diffuse along vacancies, surface sites or sublattice structures [70]. Consequently, the activity of halide ions increases under illumination due to the presence of more photogenerated holes, promoting the formation of I_2_. Then I_2_ diffuses within the perovskite film, following gradients from high to low concentrations (Figure 3b) [71]. Similarly, the illuminated side with a high oxidation rate also induces this I_2_ concentration gradient within the perovskite film (Figure 3c) [71]. Therefore, in real perovskite PV applications, active halide ions undergo complex redox reactions, which collectively contribute to the rapid collapse and degradation of perovskite structures. In addition, Maier et al. found that the concentration of created halide vacancies after I_2_ removal increases due to the interaction of I^-^ and holes, followed by an increase in electron concentration (Figure 3d–f) [72]. As a result, interstitial iodide-related species (I_2_ and I_x_^−^) can be stabilized by structural relaxation in perovskites, leading to a 100-fold enhancement of ionic conductivity [72].

**Figure 3 materials-18-04776-f003:**
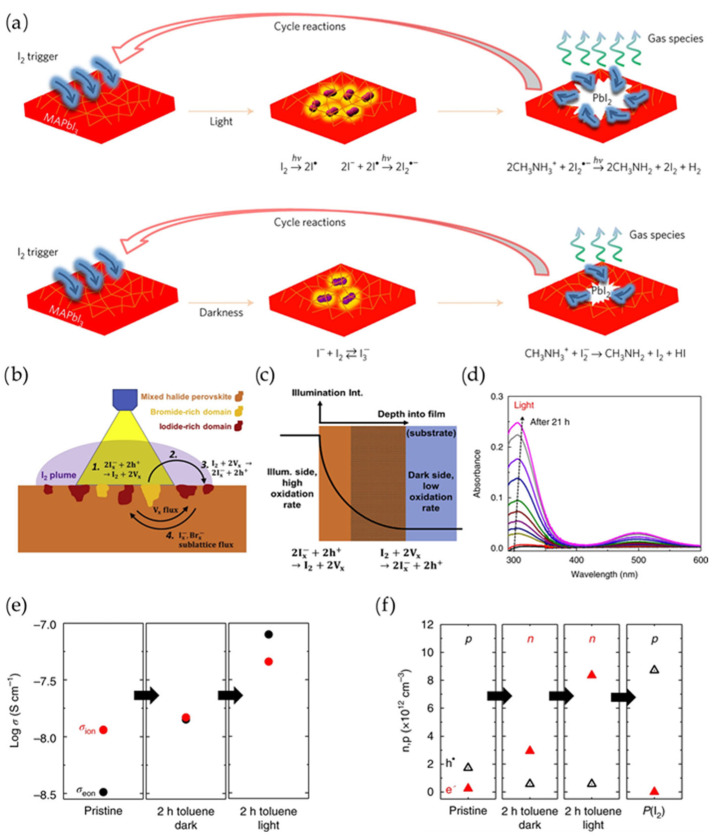
(**a**) Schematic of I_2_-induced degradation under illumination. I_2_ can further induce chemical reactions in perovskites. This was reprinted with permission from Ref. [67]. (**b**) Schematic of selective halide oxidation and I_2_ accumulation outside illuminated region with lower oxidation. (**c**) Kinetic disequilibria of iodide oxidation rate, leading to iodide gradients across the perovskite film. This was reprinted with permission from Ref. [71]. (**d**) Absorbance results as a function of time under illumination (1 mW cm^–2^). The absorption peak of I_2_ increases with prolonged illumination, as indicated by the arrow. (**e**) Conductivity variation as a function of immersion time in toluene in the dark and light conditions. (**f**) Electronic carrier concentration within perovskite film under illumination (1 mW cm^–2^) as a function of toluene treatment, reprinted with permission from Ref. [72].

More importantly, the volatile I_2_ will ultimately escape during device operation, inevitably inducing the formation of more vacancy defects [73]. These vacancies, in turn, facilitate rapid migration of adjacent iodide ions within the perovskite lattice, leading to a vicious cycle among halide vacancies, ion migration and I_2_ liberation. Furthermore, the presence of detrimental I_2_ can trigger a self-accelerating reaction with other components under illumination or applied bias conditions, leading to the rapid degradation of the operational PV (Figure 4a,b) [70]. In addition, when a perovskite-/electron-selective material (e.g., SnO_2_, TiO_2_) heterojunction is constructed, a photocatalytic process occurs under continuous illumination. This process involves the oxidation of I^-^ to I_2_ via electron transfer at the heterojunction (Figure 4c) [74]. Collectively, various complex processes contribute to the redox of halide ions under external stresses, resulting in the rapid degradation of operational perovskite PVs. Furthermore, the self-formation of molecular iodide (I_2_) in operational perovskite PVs can preferentially coordinate with I_i_ defects to form I_3_^−^ (Figure 4d) [75]. This I^3−^, as a catalyst, facilitates the self-propagating catalytic degradation of the cubic perovskite lattice (Figure 4e) [76,77]. In addition, V_i_ defects can arrest O_2_ molecules, yielding severe photogenerated superoxide (O^2−^) under light conditions [51,78]. The presence of active O^2−^ triggers the deprotonation of ammonium/amidinium cations, accelerating the collapse of the perovskite structure [79,80,81].

**Figure 4 materials-18-04776-f004:**
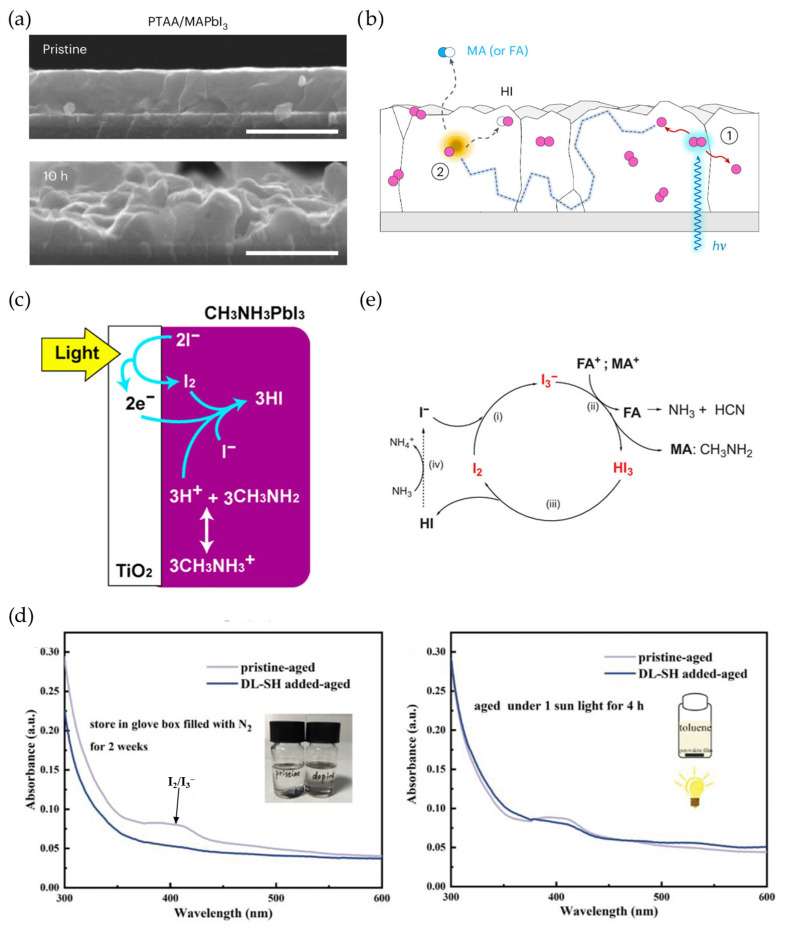
(**a**) Cross-sectional SEM images of I_2_-treated PTAA/MAPbI_3_ with and without 10 h of illumination-induced aging. Scale bars: 1 μm. (**b**) Schematic of iodine radical-induced reaction with A-site ammonium cations. Process 1 shows the release of an iodine radical from I_2_ under illumination; process 2 shows the formation of HI gas induced by iodine radicals. This was reprinted with permission from Ref. [70]. (**c**) Photocatalytic process at TiO_2_/perovskite interface. I_2_ can be formed at the interface through electron transfer into electron transport materials. This was reprinted with permission from Ref. [74]. (**d**) I_2_/I_3_^-^ formation in perovskite film after storage for 2 weeks and light soaking for 4 h, respectively, reprinted with permission from Ref. [75]. (**e**) The continuous catalytic effect of I_2_ and I_3_^−^ on the decomposition of ammonium cations, reprinted with permission from Ref. [76].

## 4. Strategies Against Operational PV Degradation

### 4.1. Stabilization of A-Site Ammonium Cation

Stable protons in A-site ammonium cations, including MA^+^, FA^+^ or larger cations for two-dimensional (2D) perovskites, are critical for enhancing the operational stability of perovskite PVs. Since the deprotonation/protonation process of ammonium cations is inherently reversible, kinetically shifting this equilibrium process toward protonation is an effective strategy to mitigate deprotonation. Nazeeruddin et al. found that the acidic 2-proton of the 1,3-bis(cyanomethyl)imidazolium cation ([Bcmim]^+^) exhibits exchange interactions with NH protons of MA^+^ (Figure 5a) [82]. This interaction inhibits the deprotonation of MA^+^ to MA^0^, which can lead to a condensation reaction, as discussed above (Figure 5b). As a result, the operational stability of perovskite modules is enhanced significantly. In addition, Lewis acid can interact with a lone pair of I^−^ molecules, preventing I^−^ from capturing the proton of MA^+^ [14].

We should further consider whether the incorporation of these A-site cations can form stable 2D perovskites, instead of 2H non-perovskites. When creating stable 2D perovskites, the incorporation of tetrahydrotriazinium cations (THTZ-H^+^) with various N-H groups can exhibit a higher number of hydrogen bonds with lead octahedra, leading to more photostable perovskite PVs (Figure 5c,d) [38]. Given the challenge of synthesizing these stable A-site cations, researchers have employed in situ proton transfer processes to directly construct a stable 2D perovskite phase on a 3D perovskite surface [25,38,83]. For instance, De Wolf et al. demonstrated that the proton transfer process between P-OH in SAM and 4-hydroxybenzylamine can form large stable ammonium cations, leading to the formation of a 2D/3D heterojunction on the bottom surface of perovskites (Figure 5e–g) [28]. However, this strategy may increase the possibility of a condensation reaction between this introduced amine and the original ammonium cations. Therefore, we believe that designing higher photo- and thermal-stable ammonium cations in perovskites is a promising strategy to enhance the device’s lifetime in real applications.

**Figure 5 materials-18-04776-f005:**
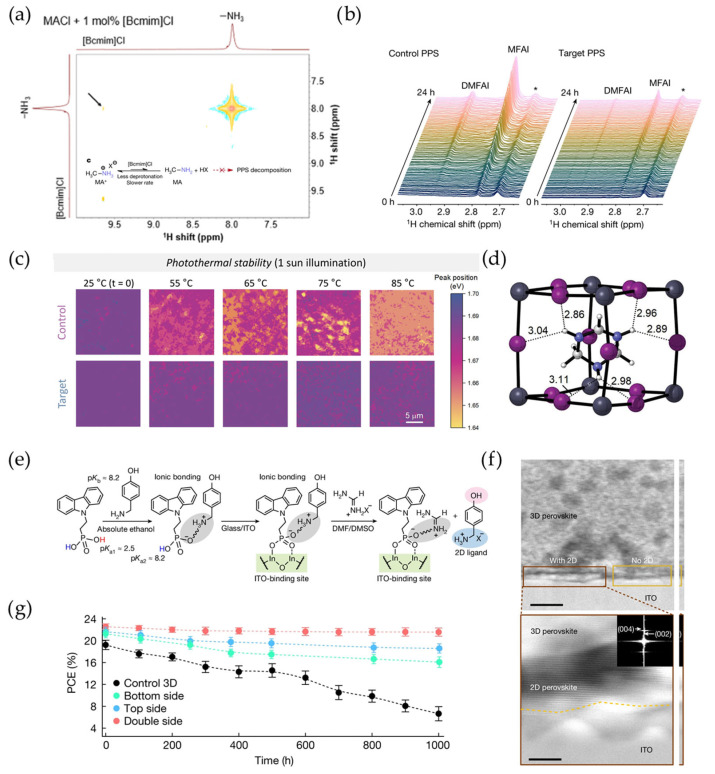
(**a**) Exchange spectroscopy of MACl + 1 mol% [Bcmim]Cl. In the inset, it is proposed that [Bcmim]Cl can stabilize the reaction to facilitate the formation of MA^+^. (**b**) 1H NMR spectra of perovskite solution degradation at 60 °C for 24 h with and without [Bcmim]Cl introduction. The peak of the side product MFAI is suppressed significantly after [Bcmim]Cl introduction. * denotes the residual solvent peak. This was reprinted with permission from Ref. [82]. (**c**) Photothermal stability tests of perovskite films with and without THTZ-H^+^ inclusion by PL mapping. We can note the uniform PL peak position maps after 30 min of continuous 1-sun illumination at various temperatures from the introduction of THTZ-H^+^ into the perovskite film. (**d**) Perovskite crystal incorporating THTZ-H^+^ with numerous hydrogen bonds, which can stabilize the perovskite structure, reprinted with permission from Ref. [38]. (**e**,**f**) The mechanism of 2D ligand formation on the ITO/2PACz surface characterized by high-angle annular dark-field STEM. Parallel 2D perovskite layers are observed on the perovskite’s bottom surface. (**g**) Operational stability of perovskite PVs at 85 °C with 2D perovskite construction on the bottom, top and both sides, respectively. This was reprinted with permission from Ref. [28].

### 4.2. Confinement of Iodine Molecules

The I_2_ molecules formed during device operation typically coordinate with I^−^ to form I_x_^−^ ions, alleviating the direct liberation of I_2_. However, I_2_ can still volatilize rapidly at elevated temperatures [84]. Therefore, the primary strategy is to delay I_2_ escape through strong interactions between additives and I_2_ or I_x_^−^. These iodine species usually accumulate at interfaces due to the rapid diffusion of I_2_ to the perovskite surface. In order to achieve this process, iodide-capturing molecules—perfluorodecyl iodide (PFI)—are introduced at the C_60_/perovskite interface to confine the escape of detrimental I_2_. The electron-withdrawing fluorocarbon chains in PFI can capture I_x_^−^ ions formed by I_2_ and mobile I^−^ at the interface (Figure 6a) [70]. As a result, device stabilities under both UV light and thermal stresses are enhanced after introducing PFI at the interface (Figure 6b). In addition, thiol copper(II) porphyrin (CuP) carrier extraction layers were designed to confine interfacial I_2_ or I_x_^−^ escape. Cu(II) ions in CuP effectively modulate the electron distribution of the porphyrin π-ring, enhancing electrostatic interactions with I_2_ or I_x_^−^ at the interface, which leads to enhanced operational device stability [85].

Introducing functional materials directly into perovskites offers another alternative approach to strongly immobilizing I_2_/I_x_^−^, including β-cyclodextrin (β-CD) [86] and bipyridine (BPy) cages with a cavity-like structure [87]. β-CD@ I_2_ exhibited a higher volatilizing temperature (160 °C) compared to free I_2_ (70 °C), indicating inhibited I_2_ release within the hydrophobic β-CD cavity (Figure 6c). The framework of the nitrogen-rich cage in BPy can coordinate with I_2_ through electron-pair interactions, effectively capturing I_2_ evaporation (Figure 6d). This strategy can trap numerous I_2_ molecules, giving them a chance to react with Pb^0^ to reform PbI_2_ (Figure 6c). Recently, Wang et al. demonstrated that a multifunctional organic framework can also be used to confine iodine, leading to significantly enhanced photothermal device stability [88]. In addition, molecules acting as reservoirs for iodine species were introduced to suppress ion migration and heal iodide-related defects in perovskites, leading to significantly enhanced perovskite PV stability (Figure 6e) [89,90].

**Figure 6 materials-18-04776-f006:**
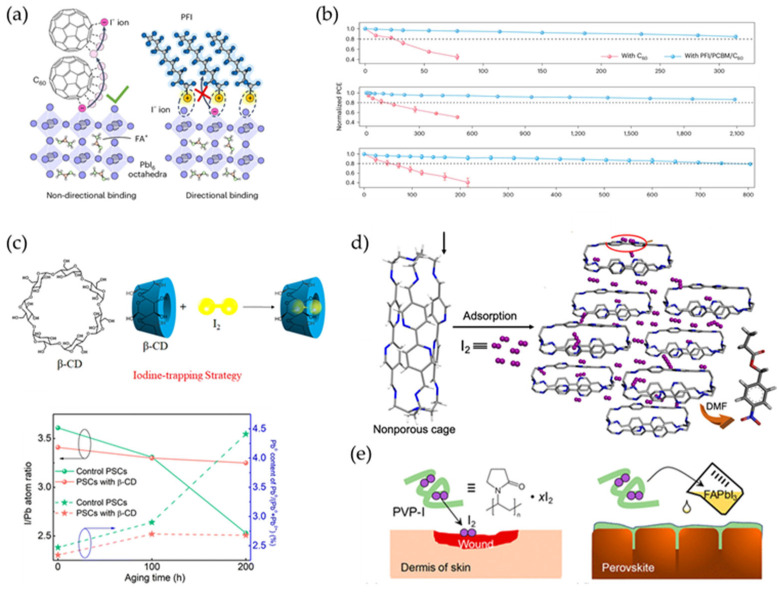
(**a**) Illustration of I_2_ capture after the introduction of PFI at the C_60_/perovskite interface. The electron-withdrawing fluorocarbon chains in PFI can coordinate with I_2_ at the interface. (**b**) Device stability tests with and without PFI intercalation at the C_60_/perovskite interface. From top to bottom: UV stability, 1-sun illumination at room temperature, and 1-sun illumination with 85 °C heating. This was reprinted with permission from Ref. [70]. (**c**) Iodine confinement by β-CD and variations in the ratios Pb^0^/(Pb^0^ + Pb^2+^) and I/Pb under 85 °C heating. I/Pb atom ratio decreases significantly without β-CD, indicating oxidation from I^−^ to volatile I_2_. This was reprinted with permission from Ref. [86]. (**d**) Illustration of iodine adsorption in BPy cage through interaction between I_2_ and nitrogen, reprinted with permission from Ref. [87]. (**e**) Schematic of the wound-healing process through the release of iodine from PVP-I, reprinted with permission from Ref. [90].

### 4.3. Reduction of Iodine Molecules

To further improve the durability of perovskite PVs, reductive additives were introduced to induce the reduction of I_2_ to I^-^ to reform a stable perovskite lattice, inhibiting ionic iodide loss during PV operation. Actually, the researchers found that the I_2_ molecules had already been generated due to the straightforward oxidation of organic halides in the aged perovskite precursor solution. This I_2_ generation can induce more halide vacancy defects in the formed perovskite film. Various types of reductants have been employed to stabilize the halide ions in the perovskite solution. Hydrazine derivatives and formate derivatives are the most commonly used reductants to reduce I_2_ to guarantee fresh organic halides [91,92,93,94]. The hydrazine group (-NHNH_2_) can reduce I_2_ as follows: -NHNH_2_+2I_2_ = -N^2+^ + 4I^−^ + 3H^+^ (Figure 7a) [75,95,96]. Similarly, the reductive reaction between formate derivatives and I_2_ is as follows: HCOO^−^ + I_2_ = CO_2_ + 2I^−^ + H^+^ (Figure 7b) [94]. Meanwhile, functional groups in these molecules can also regulate the perovskite crystallization process, further suppressing defect formation, which leads to enhanced operational stability. For instance, Xu et al. demonstrated that formate ions in dimethylammonium formate (DMAFo) can effectively reduce I_2_ to improve the long-term storage of the solution in ambient air. Furthermore, DMA counter-ions can induce the occurrence of a secondary 4H/6H phase during perovskite crystallization, leading to improved perovskite crystallinity (Figure 7c–e) [93].

**Figure 7 materials-18-04776-f007:**
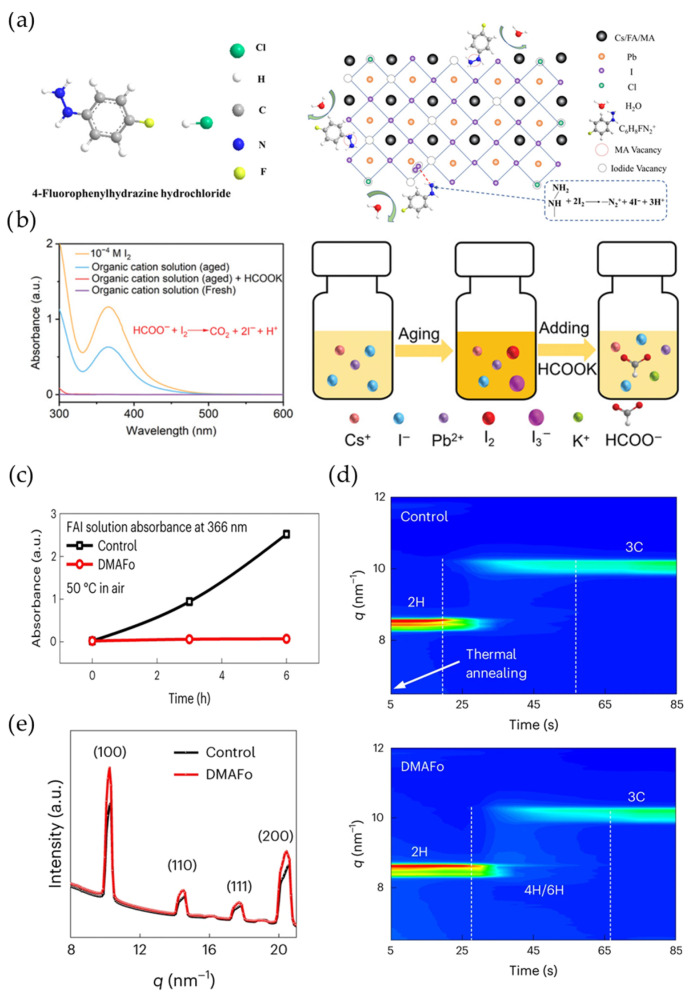
(**a**) Chemical structure of 4-fluorophenylhydrazine hydrochloride (4F-PHCl) and its reductive mechanism in Cs_0.05_(FA_0.83_MA_0.17_)_0.95_Pb(I_0.83_Br_0.17_)_3_ perovskites, reprinted with permission from Ref. [96]. (**b**) UV–vis absorption spectrum comparison of organic cation solutions with and without HCOOK addition. The peaks at 366 nm represent I_2_/I_3_^−^ formation. This was reprinted with permission from Ref. [94]. (**c**) The absorbance evolution at 366 nm of the FAI solution before and after DMAFo introduction during aging at 50 °C. (**d**) The intensity evolution of X-ray scattering (GIWAXS) at a wide grazing incidence angle along the qz direction during perovskite crystallization before and after DMAFo introduction. Secondary 4H/6H phases are formed during thermal annealing. (**e**) The integrated intensities of GIWAXS patterns for the control and DMAFo-introduced perovskite films. The DMAFo-introduced perovskite films show higher (100) and (200) peaks than the control sample, indicating improved crystallinity. This was reprinted with permission from Ref. [93].

However, operational perovskite PVs can continuously give rise to I_2_ generation, which progressively depletes the limited reductive molecules and ultimately compromises long-term device stability. Therefore, strategies of I_2_ confinement and I_2_ reduction by limited molecules cannot achieve long-term stability comparable to that of silicon PVs, which are currently limited at the laboratory scale. In order to address this issue, researchers have employed organic redox mediators to persistently convert the generated I_2_ and Pb^0^ via redox reactions. In 2019, Yan et al. applied a europium ion pair of Eu^3+^-Eu^2+^, which acts as a “redox shuttle”, to selectively reduce I^0^ and oxidize Pb^0^ defects simultaneously [97]. Alex K.-Y. Jen et al. demonstrated that their designed anthraquinone (AQ) derivative can reduce I_2_ and oxidize Pb^0^ due to its suitable redox potential, resulting in efficient electron shuttling between iodine and lead (Figure 8a) [66]. As a result, the regenerated I^−^ and Pb^2+^ reintegrate into to the perovskite lattice to stabilize the [PbI_6_]^4−^ octahedral structure. This strategy achieved significantly enhanced long-term stability of the wide-bandgap perovskite PV, retaining 95% of its initial efficiency after 500 h of operation under 1-sun illumination without UV filtering (Figure 8b). More recently, a thiol–disulfide/I^−^-I_2_ redox pair was proposed for dynamically repairing halide vacancy defects. Ge et al. utilized mercaptoethylammonium iodide (ESAI) to continuously reduce I_2_, healing iodine-induced degradations under light and thermal conditions (Figure 8c) [98]. The morphological results show that ESAI suppressed the formation of numerous pinholes (Figure 8d), which would have been generated by I_2_ volatilization during irreversible perovskite decomposition. Furthermore, Wang et al. grafted several thiol (–SH) substituent groups on the β-cyclodextrin (β-CD) cavity to obtain β-CD-(SH)_7_ (Figure 8e). These supramolecules can not only confine I_2_ (as mentioned in Section 4.2) but also enable redox cycling to enhance device durability through a thiol–disulfide/I^−^-I_2_ redox pair (Figure 8f,g) [99]. The perovskite PV demonstrated a significantly extended lifetime of T_98_ > 2780 h during device operation at 45 °C (Figure 8h). Until now, perovskite PVs have not achieved comparable lifetimes to those of silicon PVs, which is attributable to various factors rather than being solely dependent on photoactive layer optimization. We believe that the strategy of continuously reducing I_2_ back to I^−^ within the perovskite lattice shows considerable promise for industrial applications. We note that this strategy should mitigate the formation of H^+^ during reductive reactions, which can otherwise coordinate with I^−^ to form HI gas and lead to partial iodide loss during device operation.

## 5. Conclusions and Outlook

The operational stability of perovskite PVs has achieved remarkable improvement over the past several years. Differing from inorganic PV materials, perovskite materials include fragile small organic species, which exhibit complex chemical reactions under long-term illumination, heat and bias stresses. In this review, recent advances in operational stability are summarized. Then, the fundamental chemical reaction pathways of ammonium cations and mobile halide ions under various conditions are analyzed systematically. Finally, we discuss strategies targeting these chemical reactions to enhance the intrinsic operational stability of perovskite PVs. Until now, the limited stability of perovskite PVs had not yet been fully understood. Therefore, future research could focus on three aspects to guide the development of operational stability in perovskite PVs as follows (Figure 9).

(1)Vacancy formation should be suppressed to substantially inhibit mobile ion migration and I_2_ generation. Ion hopping by vacancies leads to severe ion migration. The presence of mobile ions readily induces the oxidation of I^−^ into volatile I_2_, which produces more halide vacancies, accelerating ion migration. Although various strategies have been developed to reduce ion migration and I_2_ formation, halide vacancies are the primary driver behind ion migration within perovskites [63]. Therefore, the control of vacancy formation during perovskite crystallization is essential.(2)Quantifying the relationship between mobile ion migration and perovskite PV degradation is of critical importance. Perovskite PVs show more pronounced ion movement compared to those of inorganic photoactive materials. Furthermore, these mobile ions, especially iodide ions, can trigger a series of chemical reactions, which accelerate the collapse of intrinsic perovskite lattices. Although reports claim that the suppression of ion migration leads to delayed device degradation, the number of mobile ions suppressed and their influence on device performance degradation are not clear. Therefore, it is essential to determine how the concentration of mobile ions in perovskites influences the degradation rate. We believe that device performance evolution during operation cannot follow a linear degradation trend due to the dynamic variation in mobile ion concentration. Furthermore, the increased rate of mobile ion concentration enhancement under different external stresses (e.g., illumination at different wavelengths) should be investigated systematically.(3)An accelerated aging test model for studying perovskite PV stability should be constructed. Until now, empirical insights into PV performance degradation models have largely been drawn from crystalline silicon photovoltaics [26,100]. However, the degradation of perovskite PVs associated with organic chemical reactivity differs from that of silicon PVs significantly [101], leading to more complex device degradation behavior. Therefore, the influence of organic species reactivity and halide ion migration on PV degradation pathways should be investigated, which could provide critical insights into their operational reliability under real-world conditions. In order to address the diversity of degradation mechanisms, artificial intelligence (AI), particularly machine learning and neural networks, can identify latent correlations within high-dimensional simulations and experimental datasets. Therefore, AI can predict degradation pathways and lifetimes more accurately.

## Figures and Tables

**Figure 8 materials-18-04776-f008:**
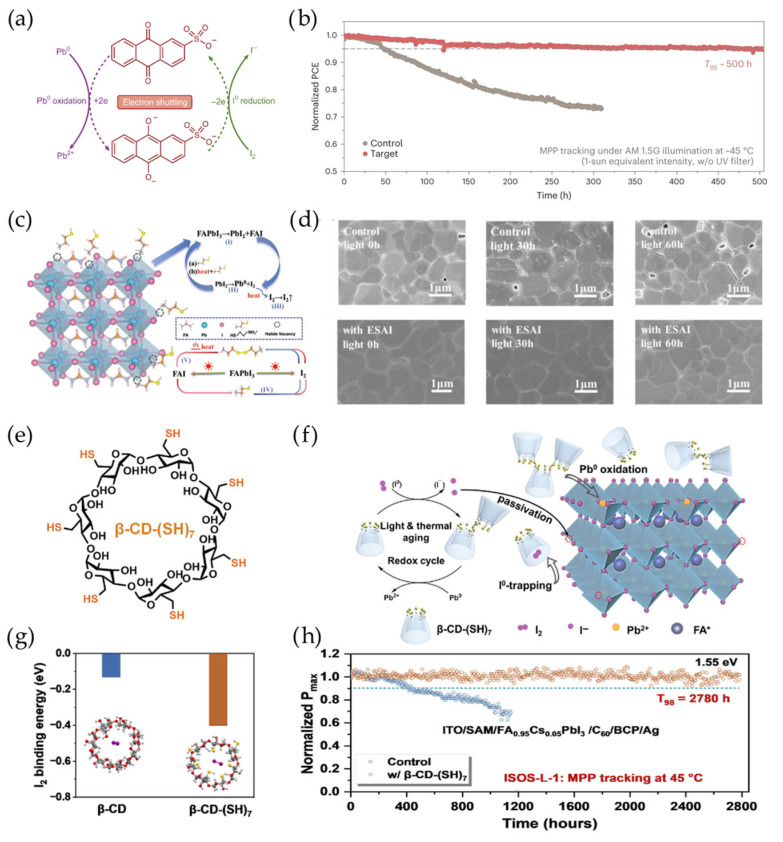
(**a**) Electron shuttling between AQ, lead and iodine. (**b**) Operational stability tests of wide-bandgap perovskite PVs without temperature control, reprinted with permission from Ref. [66]. (**c**) Light- and temperature-induced degradation mechanism before and after thiol-derivative (ESAI) introduction. I_2_ and Pb^0^ are formed through a photolysis reaction with PbI_2_ (reaction III). The thiol group in ESAI can reduce I_2_ back to I^-^, followed by the formation of a thiol–disulfide pair (reaction IV). (**d**) Scanning electron microscopy (SEM) images of the control and ESAI-treated perovskite films with different heating durations under illumination. The formation of voids is suppressed significantly by the reduction of I_2_ back to I^-^ by ESAI. This was reprinted with permission from Ref. [98]. (**e**) The molecular structure of β-CD-(SH)_7_. (**f**) The mechanism of redox cycling of lead/iodine in perovskite films by β-CD-(SH)_7_. (**g**) The binding energy between I_2_ and β-CD/β-CD-(SH)_7_. I_2_ has a higher binding energy with β-CD-(SH)_7_ compared to β-CD due to the strong interaction between I_2_ and the –SH group. (**h**) MPP stability of perovskite PV devices with and without β-CD-(SH)_7_ treatment, reprinted with permission from Ref. [99].

**Figure 9 materials-18-04776-f009:**
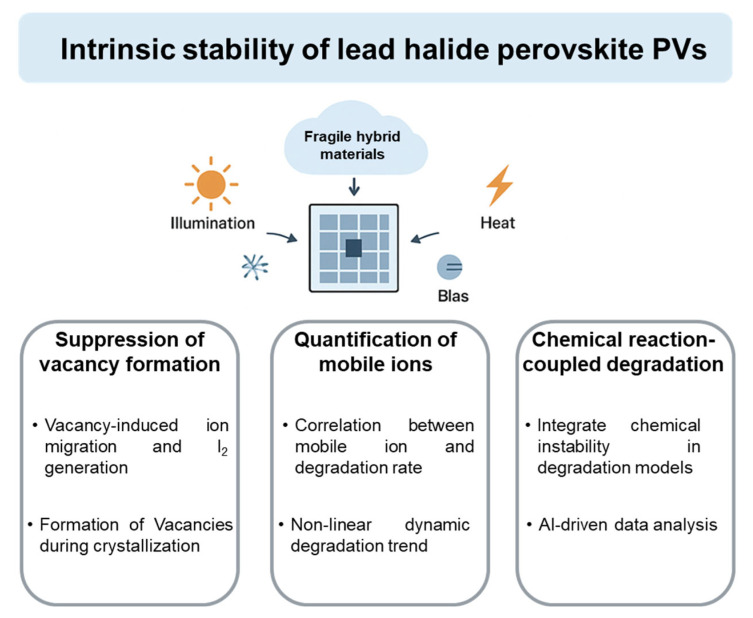
Perspectives on studying the perovskite degradation mechanism.

**Table 1 materials-18-04776-t001:** Summary of operational lifetime of perovskite PVs.

Operational Stability Test Conditions	Initial PCE	Lifetime	Device Structure	References
55 °C, 85%RH	24.7%	T_98_ = 1000 h	2PACz/FA_0.84_ Cs_0.05_MA_0.11_Pb(I_0.987_Br_0.013_)_3_ /C_60_/SnO_2_/Ag	Dong, B. et al. *Nat. Energy*. 10, 342–353 (2025) [15]
85 °C, 60%RH	23.0%	T_95_ = 2000 h	SnO_2_/FA_0.96_Cs_0.04_PbI_3_ /HTL(Unknown)/Au	Li, S. et al. *Nature*. 635, 82–88 (2024) [16]
Unknown	20.1%	T_95_ = 1600 h	SAM/FA_0.87_Cs_0.13_Pb(I_0.9_Br_0.1_)_3_ /PCBM/BCP/Cr/Au	Lin, Y.-H. et al. *Science*. 384, 767–775 (2024) [17]
65 °C, 50%RH	23.2%	T_95_ = 1200 h	SAM/Cs_0.05_MA_0.1_FA_0.85_PbI_3_/C_60_/SnO_2_/Ag	Chen, H. et al. *Science*. 384, 189–193 (2024) [18]
25 °C, 85%RH	23.8%	T_90_ = 1000 h	NiOX/FA_0.9_Cs_0.1_PbI_3_/C_60_/BCP/Ag	Chen, J. et al. *Nat. Energy*. 10, 181–190 (2025) [19]
40 °C, 45%RH	~23.5%	T_90_ = 1142 h	SAM/FA_0.85_Cs_0.05_MA_0.05_Rb_0.05_Pb (I_0.95_Br_0.05_)_3_/PCBM/BCP/Cr/Au	Li, S. et al. *Nature*. 635, 874–881 (2024) [20]
60 °C	25.1%	T_97_ = 1800 h	PTAA/FA_0.85_Cs_0.05_MA_0.1_Pb(I_0.97_Br_0.03_)_3_ /C_60_/BCP/Cu	Zhu, H. et al. *Nat. Photonics*. 1–8 (2024) [21]
45 °C, N_2_	25.95%	T_92_ = 500 h	Unknown	Li, Q. et al. *Nat. Energy*. 9, 1506–1516 (2024) [22]
45 °C	25.5%	T_88_ = 500 h	TiO_2_/(FAPbI_3_)_0.97_(MAPbBr_3_)_0.03_ /Spiro-OMeTAD/Au	Zhao, C. et al. *Nat. Commun*. 15, 7139 (2024) [23]
85 °C, 50%RH	~20.0%	T_100_ = 2000 h	NiOX/FA_0.79_Cs_0.05_MA_0.16_Pb_3_ /C_60_/BCP/Cu	Yang, Y. et al. *Nat. Energy*. 9, 37–46 (2024) [24]
65 °C, 50%RH	~22.0%	T_94_ = 1200 h	NiOX/FA_0.90_Cs_0.04_MA_0.06_Pb_3_ /C_60_/BCP/Cu	Zhu, X. et al. *Adv. Mater*. 36, 2409340 (2024) [25]
25 °C, N_2_	22.7%	T_99_ = 4500 h	TiO_2_/FAPbI_3_/PTAA/Au	Suo, J. et al. *Nat. Energy.* 9, 172–183 (2024) [26]
85 °C, 85%RH	23.2%	T_87_ = 1900 h	TiO_2_/SnO_2_/FA_0.9_Cs_0.05_MA_0.05_PbI_3_/ Spiro-OMeTAD/MoOx/ITO/Au	Ding, Y. et al. *Science*. 386, 531–538 (2024) [27]
40 °C, 50%RH	~23.0%	T_90_ = 1000 h	2PACz/FA_0.95_Cs_0.05_PbI_3_/C_60_/SnO_2_/IZO/Cu	Azmi, R. et al. *Nature*.628, 93–98 (2024) [28]
65 °C, 50%RH	23.5%	T_96_ = 2000 h	NiO_X_/FA_0.90_Cs_0.05_MA_0.05_Pb_3_ /C_60_/BCP/Ag	Liu, C. et al. *Science*. 382, 810–815 (2023) [5]

## Data Availability

No new data were created or analyzed in this study. Data sharing is not applicable to this article.

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
