# Peer review of "A Review of the Intrinsic Chemical Stability Challenge in Operational Perovskite Photovoltaics"

_materials, 2025, doi:10.3390/ma18204776_

Round 1

Reviewer 1 Report

Comments and Suggestions for Authors

The review article titled "A review of intrinsic chemical stability challenge in operational 2 perovskite photovoltaics" submitted by Zhenhua Xu et al., provides a structured summary of intrinsic chemical stability issues in perovskite photovoltaics but lacks novelty. The manuscript effectively outlines three primary intrinsic degradation mechanisms: deprotonation of A-site ammonium cations, migration of X-site halide ions, and the oxidation of halide ions. It logically follows this with a discussion of strategies to mitigate these issues. However, the report identifies minor shortcomings that must be addressed before publication.

  1. There are some spelling errors. Please fix it.

    (i) Page 1, line 23 “theses reactions”  = these reactions

    (ii) Page 4, line 97 “Specially, deprotonation” = Specifically, deprotonation

    (iii) Page 4, line 42 “organic compositions” = organic components

    (iv) Page 5, line 128 “deportation” = deprotonation

    (v) Author used both summaries and summarises (line 48, 57, 72, 370). Please use any of these ones.

  1. The introduction is compact. However, the use of citations is not appropriate for a review article introduction. As an example, “The power conversion efficiency of perovskite photovoltaics (PV) has already been 34% competitive with the crystalline silicon photovoltaics.[1]” = the ref .[1] is correlated to the Mutual stabilisation of hybrid and inorganic perovskites and so on. I suggest adding more specific reviews as citations for broader understanding. The author can use some most cited reviews in this field or find more appropriate references --

“power conversion efficiency of perovskite……” = https://doi.org/10.1021/acs.chemrev.0c00107

“additive or interface engineering strategies….” = https://doi.org/10.1039/C9TA07657C

“perovskite lattice degradation initiated by numerous ….”=  https://doi.org/10.1039/D1QM01250A,  https://doi.org/10.1021/acsami.3c09887

  1. There is no citation in this section. “However, the lifetime of these new photovoltaic technologies lags far behind silicon-wafer-based solar cells for 20-25 years, which has not fulfilled the minimum operational stability for commercialisation. Differing from silicon photovoltaics, the reactivity of organic compositions in perovskites is detrimental to the stability of perovskite PV under external stresses. Despite significant advances in perovskite photovoltaics, a fundamental understanding of chemical instability and the degradation mechanisms underlying methods for enhancing the stability of organic components remains insufficiently explored.” Multiple reviews and articles have been published related to this part. Please find the proper citation and include it.
  2. The conclusion and outlook part should be improved by adding a summarised figure.
Comments on the Quality of English Language

Fine

Reviewer 2 Report

Comments and Suggestions for Authors

Review Report

Summary of the Manuscript

The manuscript provides a comprehensive review of the intrinsic chemical stability challenges in operational perovskite photovoltaics. The authors systematically address the key degradation mechanisms, including deprotonation of A-site cations, migration of halide ions, and oxidation of halides. They further discuss the interplay of these processes and highlight the resulting structural and chemical instabilities in perovskite devices. The review concludes with mitigation strategies, such as stabilizing A-site cations, confining iodine, and introducing reductive additives. Overall, the paper offers an organized overview of the chemical pathways affecting stability and suggests directions for enhancing device durability.

Evaluation of Methodology, Analyses, and Conclusions

The paper is well-structured, moving logically from the background and current stability status to chemical degradation mechanisms and finally to strategies for improvement. The discussion is supported by a solid selection of recent literature, and the cited works are relevant and up to date. The explanations of chemical processes (deprotonation, ion migration, redox of iodide) are clear and helpful for both specialists and non-specialists.

However, while the review is strong conceptually, there are some aspects that could be improved:

  • The methodology of literature selection is not explicitly described. It would be useful for the authors to clarify the criteria used for including studies, since review readers often expect transparency about coverage and scope.
  • The analyses are primarily descriptive. The paper could be strengthened by including a more critical evaluation of conflicting results in the literature, as well as a discussion of which stabilization strategies are most promising and scalable for commercialization.
  • The conclusions provide a useful summary but could benefit from more explicit recommendations for future experimental approaches or theoretical modeling that could help overcome current bottlenecks.

Constructive Feedback and Areas for Improvement

  1. Provide a short section on literature search methodology to clarify the review scope.
  2. Add a critical comparison of different stabilization strategies (e.g., which approaches are currently limited to lab-scale vs. which show promise for industrial application).
  3. Enhance the discussion of future perspectives by suggesting specific research directions (e.g., modeling approaches for ion migration, accelerated testing protocols beyond silicon PV standards).
  4. Several figures contain very dense information (e.g., Figure 2 and Figure 4), making them somewhat difficult to interpret at first glance. Simplifying labels or enlarging key diagrams would improve readability.
  5. The legends should be more self-explanatory. In some cases, the reader must refer heavily to the main text to understand the figure. A short explanatory note within the legend would make the figures more accessible.
  6. The source of reprinted figures is clearly indicated, but in a few cases the resolution could be improved to ensure readability when published.
  7. Improve figure clarity by enlarging complex schematics and expanding legends.
  8. Proofreading is needed to fix minor language issues and formatting inconsistencies.
  9. Could the authors clarify the criteria and methodology used for selecting the studies included in this review? For instance, were only high-efficiency devices considered, and how were conflicting reports handled?
  10. Among the stabilization strategies discussed (A-site cation engineering, iodine confinement, reductive additives, etc.), which ones do the authors believe are the most promising for large-scale commercialization, and why?

Recommendation

The manuscript presents valuable insights into the intrinsic chemical stability issues of perovskite photovoltaics and is suitable for publication after minor revisions addressing the points raised above.

Reviewer 3 Report

Comments and Suggestions for Authors

In this review manuscript, the authors presented the structural and chemical evolution of perovskite PVs under operation. The authors first summarized current progress in operational stability, then discussed chemical degradation mechanisms, particularly the reactivity of A-site cations and X-site halides under light, heat, and bias stress. Finally, they outlined the stability enhancement strategies based on these degradation pathways. In its current form, the manuscript lacked the depth of analysis. The review is overly descriptive, avoids discussing contradictions, and overstates its conclusions. To be considered for publication, the authors needed to significantly expand the critical analysis and, lastly, provide a much sharper forward-looking perspective. In the current, the manuscript should not be considered for publication.

Reviewer’s Comments

(1) In this review manuscript, the authors focus on the intrinsic chemical stability issues in perovskite photovoltaics (PV), which is indeed an important subject. However, the scope of the paper does not clearly establish its novelty compared to earlier reviews. For example, Boyd et al. (Chem. Rev., 2019) and Rong et al. (Science, 2018) already provided comprehensive reviews of degradation pathways. The current review mainly summarizes existing knowledge without emphasizing what is uniquely new, underexplored, or forward-looking. To warrant publication, the authors need to clearly define how this review advances beyond earlier works.

(2) The review repeatedly explains degradation processes, ion migration, deprotonation, and halide oxidation, but the treatment was expressive rather than critical. For example, in Section 3.2, halide ion migration was explained as a contributor to instability, but the activation energy barriers reported across different compositions (e.g., FA-based, Cs/MA mixed systems) were not compared. Similarly, the influence of processing parameters such as solvent choice or annealing conditions was ignored. For a broader audience, more quantitative comparisons should be discussed.

(3) The manuscript reproduced several schematic diagrams and figures from previous studies (e.g., Figure 2b-g), yet these were presented with little critical evaluation. For example, the concept of bias-assisted charge extraction (BACE) was summarized, but the authors did not discuss its intrinsic limitations in the literature. A stronger approach would involve not only presenting such schematics but also critically assessing their assumptions, applicability, and discrepancies in different research reports.

(4) Generally, a review article must go beyond summarization by engaging with contradictions and uncertainties in the field. The current review manuscript could not include such cases. For example, while FA-rich perovskites were often claimed to enhance stability, other studies show increased susceptibility to interfacial deprotonation when paired with certain oxide electron transport layers. Similarly, the effectiveness of mixed-cation/anion strategies was presented as commonly positive, whereas some reports demonstrated accelerated halide segregation under light bias. Authors should consider such points to broaden their perspective.

(5) Based on the title “A review of intrinsic chemical stability challenge in operational perovskite photovoltaics”, the review manuscript should be more general and consider a broader range of perovskites, including perovskite oxides (R. Jaramillo, et al., APL Mater. 7, 100902 (2019)), perovskite oxysulfides (M. Sheeraz, et al., Phys. Rev. Mater. 3, 084405 (2019)), and perovskite sulfides/selenoides (A. Yaghoubi, et al., Commun. Mater. 5, 262 (2024)). To cover the broader concept based on the title, authors should consider the suggested papers to include the contribution of overall perovskites towards photovoltaic applications.  

(6) From an industrial standpoint, the authors made some sweeping claims about stability improvements without providing quantitative benchmarks relevant to commercialization. For instance, the statement that “the operational stability of perovskite PV has achieved remarkable improvement” was unclear in the absence of a direct comparison to silicon PV, which remained the industry baseline. While reported stabilities of ~2000-5000 h under laboratory conditions represent academic progress, these fall drastically short of the >20-year operational lifetime (>175,000 h) required for bankability in the photovoltaic industry. Without discussing this gap, the review risks overstating the readiness of perovskite PV for commercialization.

(7) The conclusion and outlook could not provide specific, actionable insights by suggesting concrete research directions, new experimental methodologies, or emerging material strategies. A high-quality review should served as a roadmap for the field, pointing toward next-generation strategies (e.g., multiscale modeling of degradation pathways, integration of real-time operando characterization techniques, or the use of AI-driven accelerated discovery). The current outlook was too generic to provide meaningful guidance.

Round 2

Reviewer 3 Report

Comments and Suggestions for Authors

I accept the manuscript for publication